# Effect of Intermittent Low-Pressure Radiofrequency Helium Cold Plasma Treatments on Rice Gelatinization, Fatty Acid, and Hygroscopicity

**DOI:** 10.3390/foods13071056

**Published:** 2024-03-29

**Authors:** Ziyi Cao, Xingjun Li, Hongdong Song, Yu Jie, Chang Liu

**Affiliations:** 1School of Health Science and Engineering, University of Shanghai for Science and Technology, Shanghai 200093, China; caoziyiz@163.com (Z.C.); cau4080@163.com (H.S.); 2National Engineering Research Center for Grain Storage and Transportation, Academy of National Food and Strategic Reserves Administration, Beijing 102209, China; jyu@ags.ac.cn (Y.J.); shlglc@126.com (C.L.)

**Keywords:** cold plasma, rice, helium, gelatinization parameters, fatty acid profile

## Abstract

To establish the safe and reproducible effects of cold plasma (CP) technology on food products, this study evaluated the gelatinization parameters, fatty acid profile, and hygroscopic properties of rice grains repeatedly treated with low-pressure radiofrequency (RF) helium CP (13.56 MHz, 140 Pa, 120 W-20s, 0–4 times, and 300 g sample). Compared with the untreated (zero times) sample, with an increase in CP treatment times from one to four on rice, the water contact angle and cooking time decreased, while the water absorption rate and freshness index increased, and the pH value remained unchanged. CP repeating treatments essentially had no effect on the gelatinization enthalpy, but significantly increased the peak temperature of gelatinization. From the pasting profile of rice that has been repeatedly CP treated, the peak, breakdown, and setback viscosities in flour paste decreased. CP repeating treatments on rice did not change the short-range molecular order of starch. Compared with the untreated sample, the first helium CP treatment maintained the content of C18:1n9c, C18:2n6c, and C18:3n3, but the second to fourth CP treatment significantly decreased contents of these fatty acids (FAs) as the C18:0 content increased. The first three CP treatments can increase the water and sucrose solvent retention capacity in rice flours. CP repeatedly treated rice first exhibits the similar monolayer water content and solid surface area of water sorption. Principal component analysis shows that contact angle, pasting parameters, and fatty acid profile in milled rice are quite sensitive to CP treatment. Results support that the effect of low-pressure RF 120W helium CP treatment 20 s on rice grains is perdurable, and the improvement of CP intermittent treatments on rice cooking and pasting properties is an added benefit, and the hygroscopic properties of rice was kept.

## 1. Introduction

The respect for nature, demand for healthy nutrition, and thereby green technology is increasing all over the world. Incorporating cold plasma (CP) technology into the food industry is significant because it can cut the usage of chemicals [1]. Plasma reactive species are in charge of microbial decontamination and food specific structural modifications [2,3], and they can facilitate nutrients extraction from biological systems. There is an urgent need for extensive research into the impact of CP treatment upon the quality and physico-chemical indexes of cereal grains and food.

CP can modify starch by increasing surface energy, introducing functional groups, cross-linking, and depolymerization, as well as changing hydrophilicity, or molecule degradation and crystal etching [3,4]. The impact of CP varies depending on the CP resources, components, production method, treatment time, and starch properties. There are several reports on CP modification of the rice kernel surface and improvement in the taste quality of cooked brown rice and milled rice [5,6]. There are, however, few studies that address the influence of intermittent treatment and the duration of the cold plasma effect.

Some research considered that air plasma is rich in reactive oxygen-based and nitrogen-based species [7], and the resultant pH reduction was used to explain the action mechanism of CP on food. Because HNO_x_ (x = 1, 4) can be produced in the humid air of CP when dielectric barrier discharge (DBD) CP was used to treat zein powder [8], peanut protein [9], and whey isolate protein [10], CP could decrease the pH values of these food materials. However, when the argon or oxygen cold plasma from KiNPen 09 direct current (DC) plasma jets was used to treat gellan gum, significant changes were not observed to pH and the content of titratable acid [11]. After air DBD CP treatment, the pH of the rice flours significantly increased [12]. The effect of low-pressure, RF high-purity helium CP on rice pH is worth further study.

Pal et al. [12] discovered that swelling force, light transmittance, and syneresis in short-grain and long-grain flours increased significantly with increasing air DBD CP power (60, 70 kV) and treatment time (0, 5, and 10 min), whereas blue value, final viscosity of paste, and setback viscosity decreased in short-grain rice flours but increased in long-grain rice flours. As for starch retrogradation, the intermittent treatments of helium CP on rice should be evaluated.

CP can influence the modification of starch and protein in the lipid oxidation reaction, causing lipid-containing food to deteriorate [13]. Radiofrequency (RF) CP (300 W, 13.56 MHz) treatment for 13.33 min resulted in the partial oxidation of the plant wax surface and the production of carboxylic acid and aldehydes [14]. The CP oxidation was observed in many foods; lipid oxidation should be avoided in foods with high lipid content. Removing oxygen from the feed gas and decreasing the CP treatment powder and time could reduce this oxidation [15]. The impact of high-voltage atmospheric CP (HVACP) treatment on soybean oil using hydrogen gas was mainly a reduction of polyunsaturated fatty acids and the increase in saturated fatty acids, while avoiding the formation of undesirable trans-isomers [16]. The effect of CP intermittent treatment on the fatty acid profile of cereal grain and its products needs to be studied further.

Numerous effects of CP, such as the organoleptic and biomolecular impact, plant cell structure modification, and the microbial impact, have been extensively studied [17,18,19]. However, there are still many gaps in our understanding of the impact mechanisms of CP on the organoleptic and biomolecular properties of food. These gaps should be thoroughly examined and the potential cumulative effects or alterations can be uncovered to ensure sustained benefits before this technology will be viable for food industry. After plasma treatment, the particles’ surfaces become more hydrophilic [20], and the question is whether this phenomenon tends to decay gradually during the resting period. In addition, the sorption properties of grain and food is of interest because the relationship between moisture content and water activity is essential for a design and optimization of many processes in food industry such as drying, packaging, and storage [21], and few studies deal with whether cold plasma treatment does alter the hygroscopic properties of grain kernels. The present study performed tests just after low-pressure RF helium CP treatment on short-grain milled rice and repeated the tests several times to demonstrate the changes in gelatinization temperature, fatty acid profile, and hygroscopic properties, with the aim of establishing the effective, safe, and reproducible effects of CP technology on food products, and proving the industrial applicability of this technology.

## 2. Materials and Methods

### 2.1. Plasma Apparatus and Milled Rice Sample

A cold plasma processor HD-3 N from Changzhou Hanjie Biotechnology Co., Ltd., Jiangsu province, China was used for rice treatment. The plasma reactor comprises a quart glass tube with 10 mm thickness × 300 mm height × 300 mm internal diameter [22]. The top and base plates of the reactor were made of stainless steel material. The electrodes were connected on the plates through Wilson seals. The base plate had the ports through which a gas reservoir, pirani gauge, vacuum pumps, and air admittance valves were connected. The two parallel electrodes had area of 280 mm length × 250 mm width and the distance between the electrodes was kept 30 mm in all the experiments. The electrodes were capacitively coupled to a radiofrequency (PSG-II type) power supply with a frequency of 13.56 MHz. The working power range was 0 to 1000 W. The working vacuum level was 80 to 180 Pa.

Milled rice of a short-grain variety was retrieved from a rice processing plant in Changzhou, China in December 2022. Rice samples (300 g) were spread uniformly on the mesh, which was put on a glass stand between the two electrodes. The system was first evacuated to 20 Pa using three vacuum pumps to take away any adsorbed gases or water vapors from the surface of the rice kernels. The untreated sample was also treated in vacuum before analysis. Helium was used as feed gas for plasma generation and the working pressure was then adjusted to 140 Pa using a mass flow controller. The matching network was adjusted to get the stable glow discharge.

### 2.2. The Interval Test between Cold Plasma Processing

The rice was repeatedly treated at 120 W for 20 s in the CP apparatus according to the experimental procedure in Figure 1. The untreated (CK) sample was packed into a No.9 valve bag (280 mm length × 200 mm width × 0.04 mm thickness, Apple Brand, Shanghai China Manufacturing). The CP treated rice (0 h CP) was packaged into a No.9 valve bag and No.11 valve bag (400 mm length × 280 mm width × 0.04 mm thickness), and stored at RT. After CP treatment at the 48th hour, all the samples were stored at a 4 °C refrigerator for 30 days. During the analysis, all the samples were kept in a carton box (21 × 18 × 8 cm of length × width × height) in the 4 °C refrigerator. When CP processing was complete, a portion of rice kernels was stored at a refrigerator, and ca. Then, 200 g of untreated and CP treated rice were respectively milled into rice meal at a Gaosu universal pulverizer (Kewei Yongxing Instrument Co., Ltd., Beijing, China). The pulverizer has 26,000 rpm/min of rotation rate and 100–120 mesh degree of grinding. The rice meal was maintained at −20 °C until to analyze.

### 2.3. The Hydrophilicity, Water Adsorption Rate, and Cooking Time of Milled Rice

The hydrophilicity of a rice kernel was assessed with a contact angle meter (V5, Yunfan Instrument Co., Ltd., Tianjin, China). A 2.5 μL droplet of deionized water was put on the kernel surface and the water contact angle was determined using sessile drop method. Analysis was started immediately after deposition of a single droplet of deionized water on the kernel surface. The dynamics of the droplet shape was further recorded at every 10 s by a video camera.

The water adsorption rate of rice grains was measured in 50 mL plastic centrifugal tubes. Four grams of rice (*m*_1_) was weighed into a tube, then 40 mL deionized water was added. The tubes stood still at room temperature (RT) for 4 h. After pouring the distilled water out, the rice kernels were put on filter paper and the water on kernel surface was sucked up. The rice kernels were lastly weighed as *m*_2_. The water adsorption rate of rice was computed as
(1)$$Wate\;radsorption\;rate\;(\%)=\frac{m_{2}-m_{1}}{m_{1}}\times 100$$

Rice samples (2 g) were cooked in 20 mL of deionized water in a boiling water bath. The minimum cooking time was defined through moving out a rice kernel at different time intervals during cooking process, placing it on a glass plate, and pressing it with a little long-handle spoon. The rice was cooked until no white core was visible in any kernels when pressing, following the method of Tao et al. [23].

### 2.4. The Freshness Index and pH Value of Rice

The freshness index of rice was assessed as the difference between the absorption values at 615 nm and 690 nm, as reported by Takashi et al. [24] with some modifications. A reagent solution was prepared by dissolving about 300 mg of Bromothymol blue (BTB) in 200 mL of 75% ethanol solution, then diluted to 600 mL with deionized water, and finally adjusted to pH 7.0 with 0.2% potassium hydroxide solution. One gram of rice flour was ground in a mortar and pestle with 10 mL reagent solution. After centrifugation at 4000× *g* rpm for 10 min, the absorption values of the supernatant were measured on a spectrophotometer (Shanghai Aoxi Instrument Co., Ltd., Shanghai, China). The greater the difference in D_615_–D_690_ is, the greater the freshness of the rice kernels is.

The pH value of plasma-treated rice was determined using a pH meter (S210, Mettler Seven Compact, Mettler Toledo, Zuich, Swiss). One gram (1.0000 g) of rice meal was weighed into a 50 -mL plastic centrifuge tube, then 15 mL distilled water was added and stirred with a glass rod, the pH value was measured immediately and after 4 h.

### 2.5. Gelatinization Temperature

The gelatinization temperature of the rice flours was determined with a differential scanning calorimeter (DSC; 200F3, Netzsch, Selb, Germany).The sample (5.0–5.1 mg) was weighed into an aluminum crucible and deionized water was added to produce a water/sample ratio of 2:1. The aluminum crucible was then sealed and equilibrated overnight at 4 °C. The DSC temperature was increased from 20 °C to 110 °C at the heating rate of 10 °C min^−1^.

### 2.6. Pasting Property andSolvent Retention Capacity

The pasting parameters were determined with a Rapid ViscoAnalyzer (RVA) (Model RVA Tecmaster, PerkinElmer Inc., Waltham, MA, USA), using the method of Bahrami et al. [15] with some modifications. Rice flour suspension (3.5 g of flour plus 24.5 g of deionized water) were, heated from 50 to 95 °C at 6 °C min^−1^ after equilibration at 50 °C for 1 min, then held at 95 °C for 5 min, and finally cooled to 50 °C at 6 °C min^−1^. Peak viscosity, trough viscosity, breakdown viscosity, final viscosity, setback viscosity, and pasting temperature were obtained from a pasting profile.

The solvent retention capacity (SRC) of rice flour were determined using AACCI method 56–11.02 with some modifications [25]. Four 50-mL round-bottom centrifugal tubes with the covers were weighed as W_0_, and then 5.000 ± 0.050 g of rice flour (W_1_) was weighed into each tube. At the second step, to four 50 mL sharp-bottom centrifugal tubes, 25 ± 0.05 g of distilled water, 5% lactic acid, 5% sodium bicarbonate, and 50% sucrose solution were separately added, then each reagent was, respectively poured into the rice flour-containing tube and the time was recorded. The round-bottom centrifugal tubes were covered with lids and severely vortexed for 5 s in a rotary mixer (WH-866, Taicang Hualida Laboratory Equipment Co., Ltd., China), then the rice flour was allowed to swell for 20 min. Each tube was vortexed for 5 s after 5, 10, 15, and 20 min, respectively. At the third step, after the tubes were centrifuged at 4 °C and 3052× *g* rpm for 15 min, the supernatants were slowly poured out, and the tubes were placed upside-down on tissue for 10 min. Lastly, after covering with lids, the precipitate-containing tubes were weighed as W_2_. The SRC value for each reagent was calculated using Equation (2).
(2)$$SRC=\left[\frac{W_{2}-W_{0}}{W_{1}}\times \frac{86}{100(1-M)}-1\right]\times 100\%$$
where W_0_ is the weight (g) of a round-bottom centrifugal tube plus its lid; W_1_ is the weight (g) of rice flour; W_2_ is the weight (g) of a rice flour precipitate-containing tube and its lid; and M is the moisture content of rice flour (decimal, wet basis); 86% shows the dry matter content of the reference sample with M = 0.14.

### 2.7. Fourier Transform Infrared Spectroscopy

Fourier transform infrared spectroscopy (FTIR) spectra of the samples were recorded, at 64 scans with a spectral resolution of 4 cm^−1^, in attenuated total reflection (ATR) mode on a spectrometer (Nicolet 6700 FTIR, Thermo Electron Corporation, Waltham, MA, USA). The sample spectra were recorded against a background spectrum (without a sample in place). All measurements were analyzed using OMNIC software (Version 8.2a). The dried samples were mixed with KBr, and they were ground and pressed into pellets. FTIR spectra were collected in the wave number range of 400–4000 cm^−1^.

### 2.8. Fatty Acid Profile Analysis

The fatty acid profile was determined as described by Tao et al. [23]. For lipid extraction, rice flours (300 mg) were weighed into a 10 mL plastic centrifugal tube and mixed with 1 mL hexane and 0.25 mL of 2 mol/L KOH-methanol solution for 30 s. After being extracted for 40 min in the water bath at 30 °C using a 300 W ultrasonic probe, the sample was cooled to room temperature and mixed with 0.25 mL of 2 mol/L HCl solution for 1 min. After centrifugation at 8414× *g* rpm for 10 min, the supernatants were analyzed by gas chromatography-mass spectrometry (GC-MS).

Fatty acid methyl esters (FAME) in each sample were measured with a gas chromatography system (Agilent 8890 GC) equipped with a fused-silica column (HP-5MS UI; 30 m × 0.25 mm× 0.25 μm of length × internal diameter × film thickness) coupled with a mass selective detector (MSD; 5977B, Agilent Technologies, Santa Clara, CA, USA). Samples were injected in a split mode (0.3 μL, split ratio 10) at an initial temperature of 130 °C for 3 min, the temperature was then increased at 5 °C/min to 180 °C. After 8 min, the temperature was increased from 5 °C/min to 240 °C and stood there for 12 min. Helium was used as the carrier gas (1.0 mL/min) and the solvent delay time was 1.6 min. For MS system, the temperatures of injection port, ionization source, transfer line and quadrupole were 260 °C, 230 °C, 280 °C, and 150 °C, respectively, and electron-impact mass spectra were recorded at the 70 eV ionization voltage. The acquisitions were performed in full-scan mode (40–400 amu).

The MS information of each FAME peak was submitted to MassHunter Qualitative Analysis V10.0 software (Agilent Technologies, USA) and compared with the offline MS library of NIST08s and the online NIST Chemistry Web Book, SRD 69 (https://webbook.nist.gov/chemistry/#, accessed on 1 January 2023), considering a minimum similarity value of 80%. The measured concentration, expressed in μg/L, was quantified with the external standard method, in which 37 kinds of FAME (Sigma, Albuquerque, NM, USA) mixed with different gradients were adopted. The linear ranges of the standard curves were 0.5–400 μg/L, with *R*^2^ > 0.99.

### 2.9. Moisture Sorption Isotherms and Solid Surface Area

The moisture sorption isotherms of CP treated rice was determined with dynamic water sorption analyzer (SPS11-10μ, ProUmid GmbH & Co., KG, Ulm, Germany). This instrument can automatically, gravimetrically determine the water vapor adsorption and desorption of multiple samples in a test atmosphere chamber with controlled temperature and equilibrium relative humidity (ERH). Temperature over time is ±0.1 K, ERH accuracy is ± 0.6% ERH at (23 ± 5) °C for the range of 0–100% ERH. The equilibrium moisture content (EMC) of each rice sample (*ca.* 2.0000 g) at four constant temperatures (5, 15, 25, and 35 °C) over range of 10–90% ERH were determined using deionized water producing relative humidity and high-purity nitrogen blowing dry and preventing samples from getting moldy. The interval between gravimetrical cycles was set as 10 min. The measurement cycle began at 10% ERH and at first increased with a 10% step to 20% ERH. Subsequently ERH was raised to from 30% to 80% and finally to 90% ERH. The time per cycle was set up to a minimum of 50 min and a maximum of 50 h. The default weight limit was +100%, balance band width (dm/dt) was ±0.01%/40 min. During a measurement cycle, the samples were automatically placed on an electronic balance and weighed. One sample pan remained unloaded and was always used for drift compensation of the measured values. The recorded data were shown by SPS-Toolbox Basic Rel. 1.15 software.

Experimental EMC/ERH data was used to establish isotherm curves in Kaleidagraph for Mac 4.5.2v software, with ERH and EMC data, respectively entered onto the x- and the *y*-axis. The modified Chung-pfost equation (MCPE, Equation (3)) and Guggenheim-Anderson-de Boer (GAB, Equation (4)) were employed to fit the EMC/ERH data of rice [26].
(3)$$M=-\frac{1}{C}ln\left[-\frac{\left(t+B\right)ln\left(ERH\right)}{A}\right]$$
(4)$$M=\frac{M_{m}\cdot B\cdot C\cdot ERH}{\left(1-B\cdot ERH\right)(1-B\cdot ERH+B\cdot C\cdot ERH)}$$
where *ERH* is equilibrium RH, decimal; *M* is equilibrium MC (% wet basis); *M*_m_ is monolayer water content (% dry basis); *t* is temperature (°C); A, B, and Care the equation coefficients.

Fitting was investigated by non-linear regression analysis in SPSS v17.0 for Windows [27]. The criteria used for determining the equation for the EMC/ERH data were the determination coefficient (*R*^2^), residue sum of squares (*RSS*), standard error (*SE*), and mean relative percentage error (*MRE*). Equations (5)–(8) were used to calculate *R*^2^, *RSS*, *SE*, and *MRE*, respectively.
(5)$$R^{2}=1-\sum_{i=1}^{n}{\left(m_{i}-m_{pi}\right)}^{2}/\sum_{i=1}^{n}{\left(m_{i}-m_{mi}\right)}^{2}$$
(6)$$RSS=\sum_{i=1}^{n}{\left(m_{i}-m_{pi}\right)}^{2}$$
(7)$$SE=\sqrt{\sum_{i=1}^{n}{\left(m_{i}-m_{pi}\right)}^{2}/\left(n-1\right)}$$
(8)$$MRE\%=\frac{100}{n}\sum_{i=1}^{n}\left|\frac{m_{i}-m_{pi}}{m_{i}}\right|$$
where *m*_i_ is the experimental value, *m*_pi_ is the predicted value, *m*_mi_ is the average of experimental values, and n is the number of observations. The suitability of an equation to the EMC/ERH data of rice was thought satisfactory if the MRE was below 10% [25].

The solid surface area of water sorption was determined as given by Moraes et al. [28].
(9)$$SA=M_{m}\times \frac{N_{a}\times S_{w}}{MW}=3533.3M_{m}$$
where *SA* is the surface area of water sorption (m^2^/g solid), $M_{m}$ is the mono-molecular layer water content (% d.b.), $N_{a}$ is the number of Avogadro, ${6\times 10}^{23}molecules/mole$, $S_{w}$ is the area of a water molecule ($10.6\times 10^{-20}m^{2}$), and MW is the molecular weight of water (18 g/mole).

### 2.10. Data Analysis

Except for parallel samples for the determination of EMC/ERH data, at least three replicates were tested on each rice sample for physico-chemical and rice quality parameters. SPSS software (Version 17.0 [27]) was used to analyze data. One-way analysis of variance (ANOVA) and Duncan’s new multiple-range test was used for comparing multiple of means. Statistical significance was stated at *p* < 0.05. Data reduction and factor method was used for principal components analysis and to make factional scatterplot.

## 3. Results

### 3.1. Effect of CP Repeating Treatments on the Water Absorption Rate and Freshness in Rice

Table 1 shows the effect of helium CP repeating treatments on the water absorption rate and freshness in milled rice. With an increase in CP treatment times, the water contact angle significantly decreased while the water absorption rate significantly increased. Compared with the untreated sample, CP treatment significantly decreased the cooking time, but the decrease in cooking time was less with an increase in CP treatment times. In addition, the rice freshness index increased with CP treatment times. The rice pH value remained unchanged with CP treatment times.

### 3.2. Effect of CP Repeating Treatments on Gelatinization Temperature of Rice

Table 2 shows the effect of CP repeating treatments on the gelatinization temperature of rice. CP repeating treatments significantly increased the onset (*T*_o_) and peak (*T*_p_) temperatures of gelatinization while maintaining the conclusion gelatinization temperature (*T*_c_) and the *T*_c_–*T*_o_ values, indicating that the crystallinity and cross-links in the rice surface induced by CP treatment might increase with an increase in treatment times. CP repeating treatments essentially did not change gelatinization enthalpy, but significantly increased peak enthalpy. Compared with the untreated sample, the CP treatments significantly decreased the gelatinization peak width, and all repeating treatments maintained the peak width.

### 3.3. Effect of CP Repeating Treatments on Pasting Parameters

Table 3 shows the effect of CP repeating treatments on pasting parameters of milled rice. With an increase in CP treatment times, peak viscosity (PV), breakdown viscosity (BD), and setback viscosity (SB) decreased while peak time increased, whereas trough viscosity (TV), final viscosity (FV) and pasting temperature (PT) remained unchanged. The decrease in setback viscosity can be considered to reflect the less retrogradation tendency of amylose in a starch paste. Compared with the untreated sample, 120 W helium cold plasma treatments might decrease 2.6%, 0.5%, 3.3% and 7.3% amylose retrogradation in a starch paste for once (0 h), twice (24 h), three times (48 h), and four times (30 d) treatment, respectively.

### 3.4. Effect of CP Repeating Treatments on Solvent Retention Capacity

Table 4 shows the effect of CP repeating treatments on the SRC of rice. Compared with the untreated sample, the first three times of CP treatments significantly increased water SRC and sucrose SRC, but the fourth CP treatment no longer increased these two SRC values. The first CP treatment increased NaHCO_3_ SRC, whereas the other 2–4 treatment times maintained the NaHCO_3_ SRC. Compared with the untreated sample and the samples of the first, second, fourth CP treatments, only the third CP treatment increased lactic acid SRC. These results suggest that the first three helium CP treatments might induce swelling of arabinoxylan in rice flour.

### 3.5. Effect of CP Repeating Treatments on the Deconvoluted IR Spectra

The short-range molecular order of rice starch was measured by ATR-FTIR. The ATR-FTIR spectrum of rice flours from untreated and CP repeatedly treated rice exhibited similar absorption bands (The FTIR spectra were not shown). The ratios of absorbances at 1047/1022 (R_1047/1022_) and 1022/995 cm^−1^ (R_1022/995_) acquired from deconvoluted IR spectra were given in Table 5, as the possible indicators of the degree of short-range order at the surface of starch granules. After treatments, the R_1047/1022_ and R_1022/995_ did not show significant (*p* < 0.05) changes compared with those of the CP untreated sample, regardless of the CP treatment times. The ratio of absorbances at 1068/1022 cm^−1^ (R_1068/1022_) which might show the interaction between protein and starch, did not show significant (*p* < 0.05) changes among CP untreated and treated samples.

### 3.6. Effect of CP Repeating Treatments on the Fatty Acid Profile

Table 6 shows the effect of CP repeating treatments on the fatty acid (FA) profile of rice. Nineteen fatty acids were detected in the CP repeatedly treated rice. Five species of fatty acids with a content greater than 50 μg/g were palmitic acid (C16:0), stearic acid (C18:0),oleic acid (C18:1n9c), linoleic acid (C18:2n6c), and α-linolenic acid (C18:3n3). With an increase in CP times, the contents of C6:0, C18:1n9c, C18:2n6c, C18:3n3, C20:1, and C22:6n3 decreased, contents of C10:0, C12:0, C14:0, C14:1, C15: 0, C15:1, C18:0, C18:1n9t, C20:0, and C24:1 increased, and the contents of C16: 0, C20:5n3, and C24:0 remained unchanged. These results suggest that, compared with the untreated sample, the first helium CP treatment could maintain the content of the main monounsaturated FA (C18:1n9c) and polyunsaturated FA (C18:2n6c, C18:3n3); however, the second and more times of CP treatments significantly decreased the content of these three FAs with an increase in saturated FA (C18:0) content.

Figure 2 shows the changes in the content of fatty acid components during CP repeatedly treated rice. With an increase in treatment times from one to three, total, unsaturated, monounsaturated, polyunsaturated FAs, and the ratio of unsaturated to saturated FAs (Table 4) decreased, but the fourth CP treatment significantly increased saturated FA compared with the untreated samples.

### 3.7. Effect of CP Repeating Treatments on the Hygroscopic Property of Milled Rice

Table 7 shows the parameters of MCPE fitting to the EMC/ERH data of the CP repeatedly treated milled rice. MCPE fitted well to the MC/ERH data of CP repeatedly treated milled rice due to R^2^ > 0.97 and MRE < 6.7%. Each coefficient of adsorption MCPE or desorption MCPE was similar among the CP repeating treatments. Figure 3 further confirms the same hygroscopic properties of CP repeatedly treated rice at the same temperature and relative humidity. For adsorption or desorption behavior of rice kernels, compared with the untreated samples, helium CP repeating treatments did not significantly increase the monolayer water content and solid surface area of water sorption in rice kernels (Table 8). At 25 °C and 70%ERH, the relative safe moisture content for the untreated sample, and the first, the second, the third, and the fourth CP treating samples were 14.36%, 14.49%, 14.57%, 14.49%, and 14.75% wet basis.

### 3.8. Evaluation of Effect of CP onMilled Rice Using Principal Component Analysis

For the five samples with zero to four CP treatment times, the measured fifty-one indexes were used to principal component analysis. Figure 4 shows the loading plot of principal components after varimax rotation; the five samples were gathered together, suggesting that the effect of CP on milled rice presents difference with CP treatment times, but the difference is not huge. Further make factional scatterplot of principal components of the measured fifty-one parameters (Figure 5), contact angle, pasting parameters (PV, TV, FV, BD, SB), and fatty acid profile (C16:0; C18: 0; C18: 1n9c; C18: 2n6c, total, saturated, unsaturated, monounsaturated, polyunsaturated FA) shows dispersed distribution, but other 36 parameters were gathered together. These results indicate that contact angle, pasting parameters and fatty acid profile in milled rice are quite sensitive to CP treatment.

## 4. Discussion

Clarifying the underlying physical and biochemical mechanisms of plasma-cereals interaction is crucial for maximizing desired results. In combination of low-pressure radiofrequency CP power (13.56 MHz, 140 Pa) and treatment time, our previous report showed 120 W helium CP treatment 20s on 300 g rice sample had optimal results based on the cooking properties, cooked rice texture, and rice appearance quality [20], this study further confirm how long the effect of CP treatment can be kept. In the present study, the pH value of milled rice remained unchanged to 120 W helium CP treatment times; this result is similar to the argon or oxygen cold plasma from kiNPen 09 DC plasma jet on gellan gum and the air cold plasma from AC corona discharge plasma jet on milk [29,30].The effect of CP on the pH of foods depends on food resources and cold plasma species. The Chinese scientists Yan et al. [31] has measured the emission spectra of helium plasma using dielectric barrier discharge (DBD) experiment device and found that the intensity of spectral line 3^1^P_1_→2^1^S_0_ was the strongest at a wavelength of 501.6 nm, possible indication of high density of meta-stable helium atoms. When high-purity helium was adopted in dielectric discharge apparatus of the cold plasma, the helium plasma with high-energy photons (17.7 eV) could be generated [32]. The meta-stable helium atoms and high-energy photons did not change pH values in the helium CP repeatedly treated rice, then their effect on rice thermal and thermodynamic properties were analyzed in this study.

DSC is a sensitive tool for characterizing starch retrogradation, and it gives transition temperature and enthalpy change (△*H*) in crystallite melting [33]. In the present study, the increase in *T*_p_ induced by helium CP repeating treatments on rice kernels might indicate an increase in degree of rice starch crystallinity and a possible role of CP cross-linking [34]. The gelatinization range (*T*_c_–*T*_o_) insignificantly decreased with increasing helium CP treatment times, this peak-narrowing trend may be related to an easy availability of water to rice starch induced by helium CP treatments.

RVA can give the pasting viscosities during program heating and cooling of rice flour suspension [33]. Chaiwat et al. [35] used 60 W Argon (99.999% purity) CP in a semi-continuous downer reactor to treat tapioca starch for 30 min, the peak viscosity (PV) and breakdown viscosity (BD) of starch paste decreased, but gel-holding intensity and final viscosity (FV) increased, and setback viscosity (SB) and pasting temperature remained unchanged. In the present study, with an increase in helium CP treatment times on rice kernels, peak viscosity (PV), breakdown viscosity (BD), and setback viscosity (SB) in rice flour paste decreased while peak time increased, whereas trough viscosity (TV), final viscosity (FV), and pasting temperature (PT) remained unchanged. Because setback viscosity is defined as the difference between FV and TV, its magnitude could reflect the retrogradation tendency of amylose in a starch paste [36]. With an increase in CP treatment times on rice kernels, the decrease in setback viscosity might indicate CP repeating rice kernels has a less tendency for starch retrogradation. With an increase in CP treatment times on rice kernels, an insignificant increase in pasting temperature suggests that the DSC-measured *T*_p_ is more sensitive than the pasting temperature from RVA curves.

Solvent retention capacity (SRC) could present swelling in four solvents that differ in their compatibility to three main polymers: 5% sodium bicarbonate to assess swelling due to starch damage, 5% lactic acid to assess gluten swelling, 50% sucrose to assess arabinoxylan-mediated swelling, and swelling in pure water influenced by all three components [37]. In the present study, the SRC values were determined to determine whether rice flour functional polymers are affected by helium plasma treatments. Compared with the untreated sample, the first three times of CP treatments significantly increased sucrose SRC and might induce arabinoxylan swelling.

The increase in R_1047/1022_ acquired from deconvoluted IR spectra, probably indicating the formation of relative more ordered domains in starch granules, were observed for DBD plasma or radiofrequency plasma treated starches [38]. However, Li et al. [39] observed the similar R_1047/1022_ values between DBD plasma treated chickpea starch and the native starch. The present study showed the similar R_1047/1022_ values for low-pressure radiofrequency helium plasma repeatedly treated rice and the untreated samples, and the R_1047/1022_ value for rice flour treated by the same cold plasma should be further measured.

Albertos et al. [40] showed a decrease in contents of oleic acid (C18:1, n-9) and eicosapentaenoic acid (20:5, n-3) in mackerel fillets after CP treatment. In the present study, compared with the untreated sample, following the increase in CP treatment times, C18:1n9c significantly decreased and C18:1n9t increased, but C20:5n3 remained unchanged. The active species during plasma discharge can initiate lipid peroxidation and produce hydroperoxide, which may be further converted into aldehydes or shorter-chain fatty acid compounds [41]. The helium CP repeating treatments increased the contents of decanoic acid (C10:0), lauric acid (C12:0), and myristic acid (C14:0) in rice while methyl caproate (C6:0) decreased. The formation (<18 μg/g) of trans-oleic acid in rice induced by low-pressure RF helium was significantly less than the trans-fatty acid limit (3 mg/g), stipulated by China national standard GB28050–2011 [42].

Thermodynamic properties of food can have an insight into the microstructure associated with the food–water interface, and the physical structure and theoretical surface area play roles in determining the water-binding properties of a particular product [26]. We measured the solid surface area of rice water sorption to explore this relationship. In the range of 5 to 35 °C and 10–90% ERH, the monolayer water content and solid surface area of water sorption were 7.02% d.b. and 266.7 m^2^/g solid for adsorption, and 9.46% d.b. and 369.3m^2^/g solid for desorption, respectively. The helium CP repeating treatments did not significantly increase these values. It is very interesting that helium CP could increase the hydrophilic property and water absorption rate of rice kernels through its etching action, but it kept the monolayer water content and solid surface area of water sorption. In the measured fifty-one parameters of the present study, principal component analysis shows that water contact angle, pasting parameters and fatty acid profile in milled rice are quite sensitive to CP repeated treatment. The interaction of reactive plasma species with rice surface is a complex process that contains mass and energy transfer. Starch as a sugar polymer is difficult to be polarized, but acidic or basic amino acids in proteins are easily polarized. Further research will determine the protein conformational changes and amino acid composition in helium CP treated rice.

## 5. Conclusions

The intermittent one to four treatments of low-pressure RF helium CP on rice kernels increased hydrophilicity, water absorption rate, freshness index, and gelatinization peak temperature, but they maintained pH value, gelatinization enthalpy, FTIR spectrum, and hygroscopic properties, and decreased rice cooking time and peak, breakdown, and setback viscosity of rice flour paste. Contents of the main monounsaturated FA (C18:1n9c) and polyunsaturated FA (C18:2n6c, C18:3n3) in rice kernels were kept by the first helium CP treatment but were reduced by more CP treatments whilst the content ratio of unsaturated to saturated fatty acids decreased. The first three helium CP treatments can increase water and sucrose SRC values. The principal component analysis shows that water contact angle, pasting parameters and fatty acid profile in milled rice are quite sensitive to CP repeated treatment. It is concluded that the effect of low-pressure RF helium CP treatment on rice kernels is perdurable, and the improvement of helium CP repeating treatments on rice cooking and pasting properties is additive. The helium CP could increase the hydrophilic property and water absorption rate of rice kernels through its etching action, but it kept the monolayer water content and solid surface area of water sorption. This study is useful for using CP treatments to improve rice quality.

## Figures and Tables

**Figure 1 foods-13-01056-f001:**
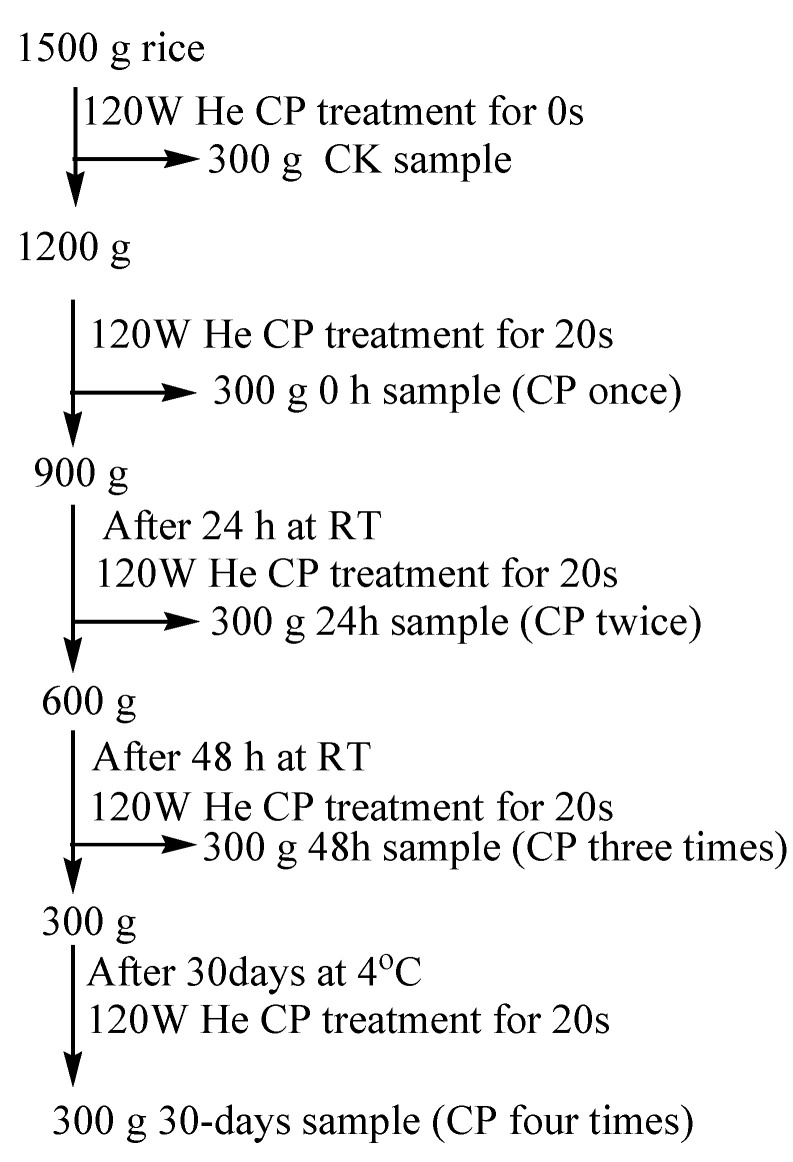
The rice sample preparations through CP intermittent treatments. Notes: He, Helium; CK, the untreated sample; CP, cold plasma; RT, room temperature.

**Figure 2 foods-13-01056-f002:**
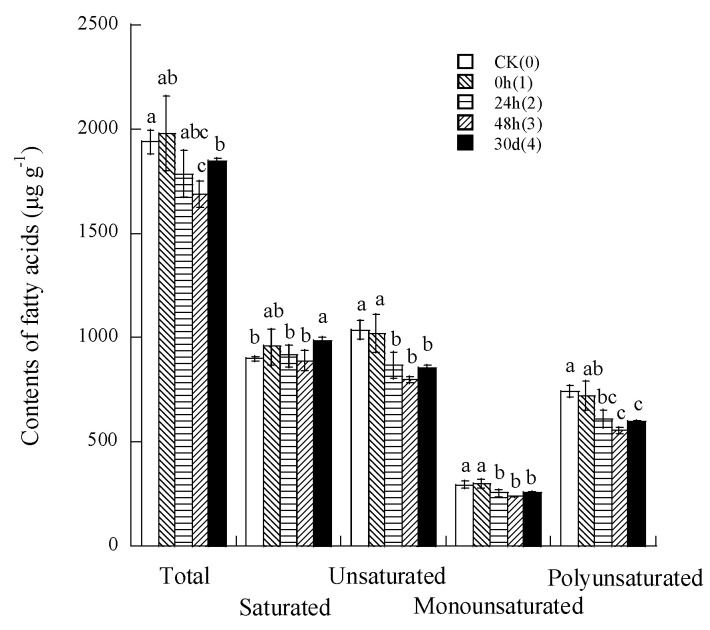
Changes in contents of fatty acid components during CP repeatedly treated rice. **Notes:** The different small letters for fatty acids species are different significantly (*p* < 0.05) among different CP treatment times.

**Figure 3 foods-13-01056-f003:**
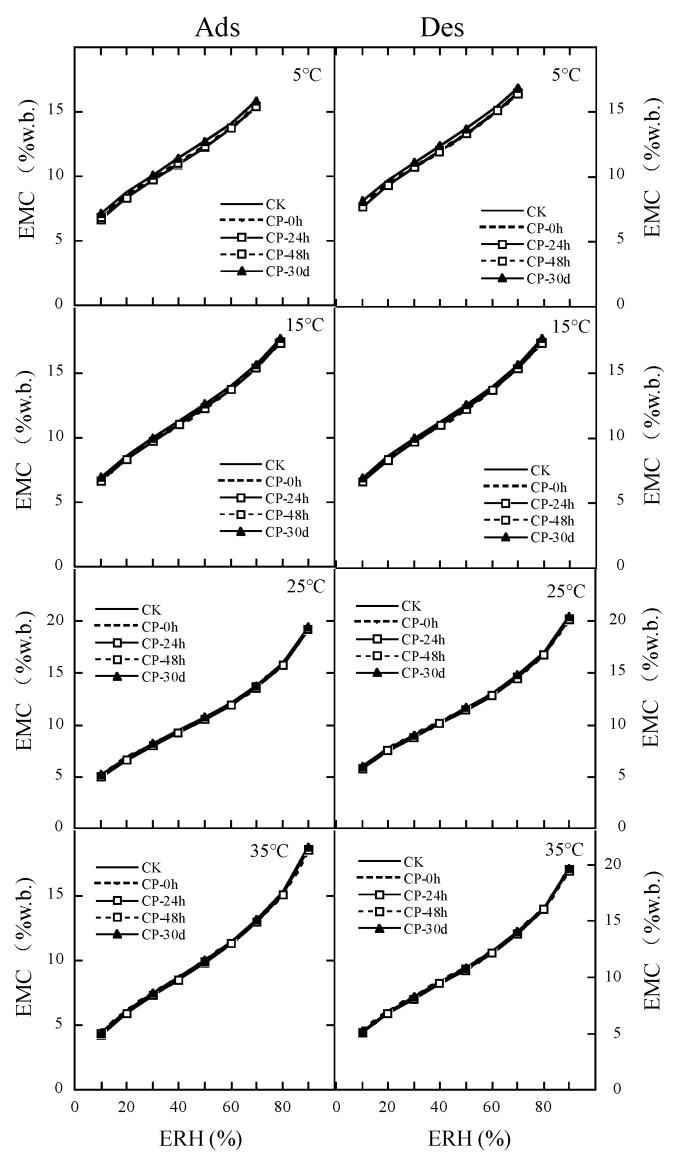
Effect of CP repeating treatments on the MCPE fitting moisture sorption isotherms of rice.

**Figure 4 foods-13-01056-f004:**
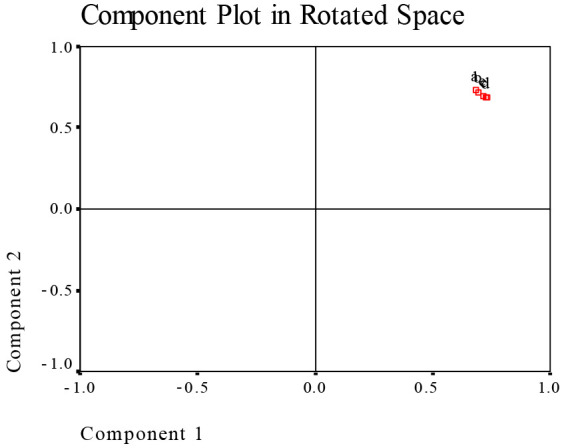
The loading plot of principal components after varimax rotation. Note: a, CK sample; b, 0 h sample; c, 24 h sample; d, 48 h sample; e, 30 d sample.

**Figure 5 foods-13-01056-f005:**
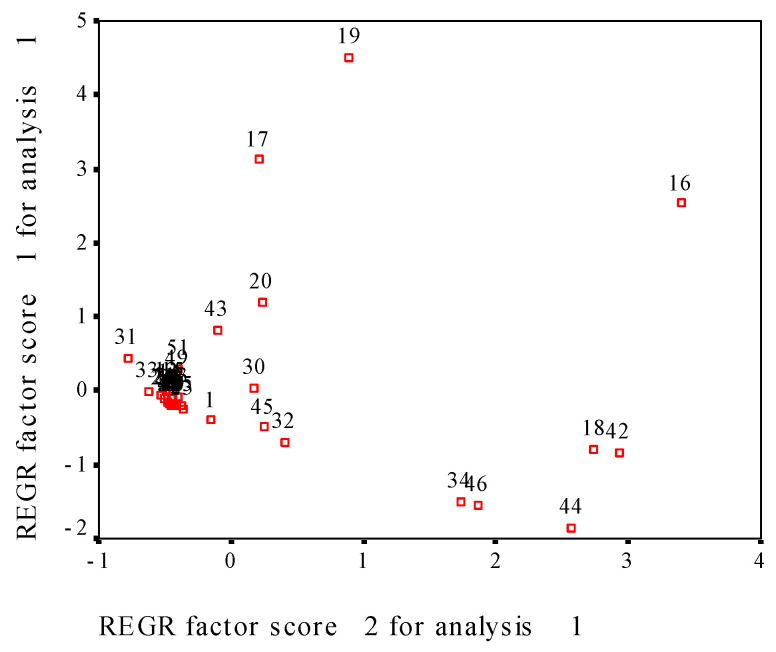
The factional scatterplot of principal components of the measured fifty-one parameters in this study. Notes: 1,water contact angle; 2, cooking time; 3, water absorption rate; 4, freshness index; 5, pH value; 6, ΔH; 7, T_p_; 8, T_o_; 9, T_c_; 10, peak width; 11, peak enthalpy; 12, water SRC; 13, NaHCO_3_ SRC; 14, Lactic acid SRC; 15, sucrose SRC; 16, PV; 17, TV; 18, BD; 19, FV; 20, SB; 21, peak time;22, PT; 23, C6:0; 24, C10:0; 25, C12:0; 26, C14:0; 27, C14:1; 28, C15:0; 29, C15:1; 30, C16:0; 31, C18:0; 32, C18: 1n9c; 33, C18:1n9t; 34, C18: 2n6c; 35, C18:3n3; 36, C20: 0; 37, C20:1; 38, C20:5n3; 39, C22:6n3; 40, C24: 0; 41, C24:1; 42, Total FA; 43, saturated FA; 44, unsaturated FA; 45, mono-unsaturated FA; 46, polyunsaturated FA; 47, ratio of unsaturated to saturated FA; 48, Mm-ads; 49, SA-des; 50, Mm-ads; 51, SA-des.

**Table 1 foods-13-01056-t001:** Effect of CP repeating treatments on water absorption rate and freshness index in milled rice.

CP Treatments	Water Contact Angle (°)	Water Absorption Rate (%)	Cooking Time (min)	Freshness Index (0.01)	pH Value
CK (0)	79.7 ± 2.90 ^a^	29.1 ± 1.39 ^d^	20.0 ± 0.52 ^a^	3.79 ± 0.36 ^d^	6.59 ± 0.03 ^a^
0 h (1)	73.8 ± 1.98 ^b^	30.9 ± 0.19 ^c^	14.5 ± 0.42 ^c^	4.75 ± 0.73 ^cd^	6.61 ± 0.03 ^a^
24 h (2)	60.8 ± 1.43 ^c^	36.6 ± 1.40 ^a^	15.5 ± 0.43 ^b^	5.59 ± 0.05 ^b^	6.59 ± 0.03 ^a^
48 h (3)	55.2 ± 4.42 ^cd^	34.8 ± 0.20 ^b^	15.0 ± 0.50 ^bc^	4.51 ± 0.04 ^c^	6.57 ± 0.01 ^a^
30 d (4)	53.6 ± 1.94 ^d^	37.1 ± 0.45 ^a^	15.5 ± 0.40 ^b^	10.25 ± 2.06 ^a^	6.57 ± 0.02 ^a^

Notes: CP, cold plasma; CK, the untreated sample; The number in bracket shows the CP treatment times; Data are expressed as mean± standard deviation (SD), number of repetitions—*n* = 3. Means with the different superscript letters (a, b, c, d) in a column are different significantly (*p* < 0.05) among different CP treatment times.

**Table 2 foods-13-01056-t002:** Effect of CP repeating treatments on the gelatinization temperature of rice.

CP Treatments	Δ*H* (J/g)	*T*_o_ (°C)	*T*_p_ (°C)	*T*_c_ (°C)	*T*_c_–*T*_o_ (°C)	Peak Width(°C)	Peak Enthalpy (0.01 mW/mg)
CK (0)	1.94 ± 0.00 ^a^	62.13 ± 0.15 ^c^	66.47 ± 0.32 ^cd^	70.50 ± 0.44 ^a^	8.37 ± 0.50 ^a^	5.30 ± 0.00 ^a^	5.46 ± 0.02 ^c^
0 h (1)	1.82 ± 0.01 ^b^	62.37 ± 0.31 ^bc^	66.53 ± 0.06 ^d^	70.33 ± 0.86 ^a^	7.97 ± 0.76 ^a^	5.00 ± 0.10 ^b^	5.47 ± 0.07 ^c^
24 h (2)	1.82 ± 0.05 ^b^	63.03 ± 0.42 ^ab^	66.83 ± 0.15 ^bc^	70.53 ± 1.19 ^a^	7.50 ± 1.05 ^a^	4.83 ± 0.21 ^b^	5.65 ± 0.18 ^bc^
48 h (3)	1.93 ± 0.03 ^a^	62.93 ± 0.38 ^ab^	67.07 ± 0.12 ^b^	70.73 ± 0.81 ^a^	7.80 ± 0.52 ^a^	4.97 ± 0.06 ^b^	5.83 ± 0.10 ^ab^
30 d (4)	1.93 ± 0.02 ^a^	63.77 ± 0.49 ^a^	67.53 ± 0.15 ^a^	71.07 ± 0.84 ^a^	7.30 ± 1.11 ^a^	4.77 ± 0.15 ^b^	6.04 ± 0.20 ^a^

Notes: CP, cold plasma; CK, the untreated sample; The number in bracket shows the CP treatment times; Δ*H*, enthalpy of gelatinization; *T*_o_, the onset temperature of gelatinization; *T*_p_, the peak temperature of gelatinization; *T*_c_, the conclusion temperature of gelatinization; *T*_c_–*T*_o_, gelatinization temperature range. Data are expressed as mean ± standard deviation (SD), number of repetitions—*n* = 3. Means with the different superscript letters in a column are different significantly (*p* < 0.05) among different CP treatment times.

**Table 3 foods-13-01056-t003:** The pasting parameters of CP repeatedly treated rice.

CP Treatments	PV (cp)	TV (cp)	BD (cp)	FV (cp)	SB (cp)	Peak Time (min)	PT (°C)
CK	4597 ± 32 ^a^	2661 ± 91 ^ab^	1937 ± 72 ^a^	4106 ± 93 ^ab^	1445 ± 7 ^a^	5.89 ± 0.03 ^c^	70.93 ± 0.06 ^a^
0 h	4595 ± 5 ^a^	2783 ± 45 ^a^	1812 ± 48 ^b^	4189 ± 29 ^a^	1407 ± 17 ^b^	5.98 ± 0.04 ^b^	70.65 ± 0.43 ^a^
24 h	4420 ± 34 ^b^	2729 ± 27 ^a^	1691 ± 29 ^c^	4165 ± 41 ^ab^	1437 ± 62 ^ab^	6.05 ± 0.04 ^ab^	71.22 ± 0.55 ^a^
48 h	4485 ± 32 ^b^	2738 ± 38 ^a^	1748 ± 57 ^bc^	4136 ± 40 ^a^	1398 ± 3 ^b^	6.02 ± 0.04 ^ab^	71.48 ± 0.51 ^a^
30 d	4220 ± 54 ^c^	2625 ± 64 ^b^	1595 ± 16 ^d^	3965 ± 59 ^b^	1340 ± 7 ^c^	6.07 ± 0.01 ^a^	71.17 ± 0.46 ^a^

Notes: CP, cold plasma; CK, the untreated sample; The number in bracket shows the CP treatment times; PV, Peak viscosity; TV, trough viscosity; BD, breakdown viscosity; FV, final viscosity; SB, setback viscosity; PT, pasting temperature. These parameters were evaluated from the pasting profile. Data are expressed as mean ± standard deviation (SD), number of repetitions—*n* = 3. Means with the different superscript letters in a column are different significantly (*p* < 0.05) among different CP treatment times.

**Table 4 foods-13-01056-t004:** Effect of CP repeating treatments on the values of solvent retention capacity (SRC) in rice.

CP Treatments	Water SRC (%)	NaHCO_3_ SRC (%)	Lactic Acid SRC (%)	Sucrose SRC (%)
CK (0)	105.95 ± 0.90 ^c^	113.66 ± 1.72 ^b^	110.47 ± 1.26 ^b^	131.25 ± 0.67 ^c^
0 h (1)	110.09 ± 1.40 ^b^	118.90 ± 0.88 ^a^	108.01 ± 1.91 ^b^	135.33 ± 0.94 ^a^
24 h (2)	110.97 ± 1.71 ^b^	112.50 ± 1.80 ^b^	110.07 ± 0.77 ^b^	132.77 ± 0.47 ^b^
48 h (3)	114.77 ± 0.90 ^a^	114.98 ± 1.32 ^b^	115.43 ± 1.60 ^a^	135.71 ± 0.06 ^a^
30 d (4)	106.65 ± 0.91 ^c^	114.68 ± 1.67 ^b^	109.77 ± 0.16 ^b^	131.88 ± 0.40 ^c^

Notes: CP, cold plasma; CK, the untreated sample; The number in bracket shows the CP treatment times; SRC, solvent retention capacity. Data are expressed as mean ± standard deviation (SD), number of repetitions—*n* = 3. Means with the different superscript letters in a column are different significantly (*p* < 0.05) among different CP treatment times.

**Table 5 foods-13-01056-t005:** IR ratios acquired from deconvoluted FTIR spectra.

CP Treatments	IR Ratios		
	R_1022/995_	R_1047/1022_	R_1068/1022_
CK (0)	1.056 ± 0.008 ^a^	0.910 ± 0.006 ^a^	0.759 ± 0.011 ^a^
0 h (1)	1.054 ± 0.021 ^a^	0.914 ± 0.008 ^a^	0.762 ± 0.012 ^a^
24 h (2)	1.065 ± 0.012 ^a^	0.904 ± 0.010 ^a^	0.739 ± 0.017 ^a^
48 h (3)	1.045 ± 0.013 ^a^	0.916 ± 0.007 ^a^	0.769 ± 0.013 ^a^
30 d (4)	1.059 ± 0.014 ^a^	0.912 ± 0.012 ^a^	0.760 ± 0.014 ^a^

Notes: CP, cold plasma; CK, the untreated sample; The number in bracket shows the CP treatment times. Data are expressed as mean ± standard deviation (SD), number of repetitions—*n* = 3. Means with the different superscript letters in a column are different significantly (*p* < 0.05) among different CP treatment times.

**Table 6 foods-13-01056-t006:** Effect of CP repeatedly treatments on fatty acid profile in rice.(μg/g).

CP Treatments	C6:0	C10:0	C12:0	C14:0	C14:1
CK (0)	33.0 ± 1.0 ^a^	26.8 ± 0.8 ^b^	27.7 ± 0.2 ^c^	22.1 ± 0.2 ^b^	3.8 ± 0.1 ^c^
0 h (1)	32.7 ± 1.5 ^a^	27.4 ± 0.2 ^b^	29.9 ± 2.5 ^bc^	23.6 ± 3.6 ^ab^	4.3 ± 0.3 ^ab^
24 h (2)	28.4 ± 0. 6 ^c^	27.7 ± 4.1 ^b^	27.5 ± 1.5 ^bc^	21.7 ± 2.1 ^ab^	4.1 ± 0.2 ^b^
48 h (3)	24.4 ± 1.2 ^d^	32.1 ± 4.9 ^ab^	33.9 ± 5.8 ^ab^	20.8 ± 1.6 ^b^	4.6 ± 0.4 ^ab^
30 d (4)	30.7 ± 0. 4 ^b^	34.9 ± 2.1 ^a^	39.2 ± 1.1 ^a^	23.2 ± 0.6 ^a^	4.8 ± 0.3 ^a^
**CP Treatments**	**C15:0**	**C15:1**	**C16:0**	**C18:0**	**C18:1n9c**
CK (0)	4.6 ± 0.2 ^c^	6.1 ± 0.4 ^b^	585.3 ± 11.2 ^a^	175.3 ± 2.5 ^c^	271.1 ± 17.3 ^a^
0 h (1)	5.3 ± 0.3 ^ab^	6.4 ± 0.3 ^b^	613.6 ± 60.1 ^ab^	197.1 ± 19.1 ^b^	267.7 ± 23.2 ^a^
24 h (2)	4.8 ± 0.6 ^bc^	6.3 ± 0.2 ^b^	573.1 ± 34.1 ^ab^	208.1 ± 9.7 ^b^	222.1 ± 15.9 ^bc^
48 h (3)	5.7 ± 0.1 ^a^	6.7 ± 0.4 ^ab^	543.2 ± 22.2 ^b^	205.9 ± 10.2 ^b^	204.5 ± 3.1 ^c^
30 d (4)	5.8 ± 0.3 ^a^	7.1 ± 0.2 ^a^	594.8 ± 5.9 ^a^	231.7 ± 4.1 ^a^	217.5 ± 3.1 ^b^
**CP Treatments**	**C18:1n9t**	**C18:2n6c**	**C18:3n3**	**C20:0**	**C20:1**
CK (0)	0.1 ± 0.1^e^	677.2 ± 28.7 ^a^	60.2 ± 0.5 ^a^	11.4 ± 0.3 ^c^	2.7 ± 0.4 ^a^
0 h (1)	9.5 ± 0.7 ^d^	657.1 ± 64.1 ^a^	59.4 ± 3.1 ^ab^	12.7 ± 0.7 ^ab^	1.6 ± 0.2 ^b^
24 h (2)	14.2 ± 0.4 ^c^	551.3 ± 41.4 ^b^	55.3 ± 1.6 ^bc^	11.1 ± 0.4 ^c^	0.5 ± 0.1 ^c^
48 h (3)	15.6 ± 0.6 ^b^	498.4 ± 13.1 ^b^	54.1 ± 0.6 ^c^	11.1 ± 1.2 ^bc^	1.3 ± 0.3 ^b^
30 d (4)	17.7 ± 1.1 ^a^	548.3 ± 4.9 ^b^	55.1 ± 0.3 ^c^	13.5 ± 0.1 ^a^	0.1 ± 0.1 ^d^
**CP Treatments**	**C20:5n3**	**C22:6n3**	**C24:0**	**C24:1**	**Ratio**
CK (0)	2.5 ± 0.1 ^ab^	5.2 ± 0.2 ^a^	14.1 ± 0.8 ^a^	9.1 ± 0.1 ^bc^	1.15 ± 0.04 ^a^
0 h (1)	2.6 ± 0.2 ^ab^	3.7 ± 0.3 ^bc^	12.9 ± 0.7 ^ab^	9.5 ± 0.4 ^b^	1.07 ± 0.01 ^b^
24 h (2)	2.3 ± 0.1 ^b^	3.1 ± 0.2 ^d^	13.1 ± 1.4 ^ab^	8.7 ± 0.3 ^c^	0.95 ± 0.95 ^c^
48 h (3)	2.4 ± 0.1 ^ab^	3.2 ± 0.6 ^cd^	12.1 ± 0.7 ^b^	9.7 ± 0.5 ^ab^	0.91 ± 0.03 ^cd^
30 d (4)	2.7 ± 0.2 ^a^	4.1 ± 0.1 ^b^	13.1 ± 0.6 ^ab^	10.3 ± 0.1 ^a^	0.87 ± 0.02 ^d^

Notes: CP, cold plasma; CK, the untreated sample; The number in bracket shows the CP treatment times; C6:0, Methyl caproate; C10:0, Decanoic acid; C12:0, Lauric acid; C14:0, Myristic acid; C14:1, Myristoleic acid; C15:0, Pentadecanoic acid; C15:1, *cis*-10-Pentadecenoic acid; C16:0, Palmitic acid;C18:0, Stearic acid; C18:1n9c, Oleic acid; C18:1n9t, Elaidic acid; C18:2n6c, Linoleic acid; C18:3n3, α-Linolenic acid; C20:0, Arachidic acid; C20:1, *cis*-11-Eicosenoic acid; C20:5n3, *cis*-5,8,11,14,17-Eicosapentataenoic acid; C22:6n3, *cis*-4,7,10,13,16,19-Docosahexaenoic acid; C24:0, Lignoceric acid; C24:1, Nervonic acid. FA, fatty acids; Ratio is the content ratio of unsaturated to saturated fatty acids. Data are expressed as mean± standard deviation (SD), number of repetitions—*n* = 3. Means with the different superscript letters in a column are different significantly (*p* < 0.05) among different CP treatment times.

**Table 7 foods-13-01056-t007:** Effect of CP repeating treatments on the moisture sorption equation Modified Chung-pfost (MCPE) in milled rice.

Sorption	Cp Treatments	Mcpe	Coefficients		Statistic	Parameters		
		A	B	C	RSS	SE	R^2^	MRE (%)
Ads.	CK (0)	439.522 ± 49.717 ^a^	38.236 ± 6.122 ^a^	0.221 ± 0.006 ^a^	18.4802	0.4620	0.9747	6.0734
	0 h (1)	449.323 ± 49.701 ^a^	39.451 ± 6.127 ^a^	0.219 ± 0.005 ^a^	17.3578	0.4339	0.9766	6.2175
	24 h (2)	426.744 ± 46.921 ^a^	38.849 ± 6.041 ^a^	0.215 ± 0.005 ^a^	18.1666	0.4542	0.9763	6.6199
	48 h (3)	427.376 ± 46.286 ^a^	37.201 ± 5.739 ^a^	0.218 ± 0.005 ^a^	17.9984	0.4499	0.9761	6.3093
	30 d (4)	414.708 ± 41.339 ^a^	33.343 ± 4.846 ^a^	0.216 ± 0.005 ^a^	17.2339	0.4309	0.9774	6.1359
Des.	CK (0)	482.952 ± 41.797 ^a^	35.951 ± 4.389 ^a^	0.216 ± 0.004 ^a^	12.0133	0.2930	0.9834	3.6602
	0 h (1)	502.329 ± 44.104 ^a^	36.613 ± 4.502 ^a^	0.216 ± 0.004 ^a^	12.0893	0.2949	0.9834	3.6179
	24 h (2)	482.612 ± 43.177 ^a^	35.693 ± 4.507 ^a^	0.213 ± 0.004 ^a^	13.2659	0.3236	0.9822	3.7853
	48 h (3)	481.107 ± 41.317 ^a^	34.809 ± 4.236 ^a^	0.215 ± 0.004 ^a^	12.3237	0.3006	0.9833	3.6594
	30 d (4)	466.352 ± 36.156 ^a^	30.724 ± 3.457 ^a^	0.214 ± 0.004 ^a^	11.3338	0.2763	0.9848	3.549

Notes: CP, cold plasma; CK, the untreated sample; The number in bracket shows the CP treatment times; A, B, and C are the MCPE constants. Ads., adsorption; Des., desorption; *RSS*, residue sum of squares; *SE*, standard error; *R^2^*, coefficient of determination; *MRE*, the mean relative percentage error. For moisture adsorptive or desorptive equation, A, B, and C are expressed as mean ± standard deviation (SD), number of observations—*n* = 34, the means with the different superscript letters in a column are different significantly (*p* < 0.05) among different CP treatment times.

**Table 8 foods-13-01056-t008:** Effect of helium CP repeating treatments on the monolayer water content (*M*_m_) and surface area values of the granular solids of rice kernels.

CP Treatment (Times)	Adsorption		Desorption	
	*M*_m_ (% Dry Basis)	Surface Area (m^2^/g Solid)	*M*_m_ (% Dry Basis)	Surface Area (m^2^/g Solid)
CK	7.02 ± 0.51 ^a^	266.7 ± 18.0 ^a^	9.46 ± 0.82 ^a^	369.3 ± 29.4 ^a^
0 h (1)	7.13 ± 0.50 ^a^	271.1 ± 17.9 ^a^	9.59 ± 0.81 ^a^	374.9 ± 28.9 ^a^
24 h (2)	7.17 ± 0.53 ^a^	272.8 ± 18.8 ^a^	9.68 ± 0.86 ^a^	378.7 ± 30.6 ^a^
48 h (3)	7.05 ± 0.50 ^a^	267.9 ± 17.9 ^a^	9.49 ± 0.53 ^a^	370.9 ± 29.4 ^a^
30 d (4)	7.23 ± 0.52 ^a^	275.2 ± 18.6 ^a^	9.76 ± 0.88 ^a^	382.1 ± 31.3 ^a^

Notes: CK, the untreated sample; CP, cold plasma; The number in bracket shows the CP treatment times; Data are expressed as mean ± standard deviation (SD), number of observations—*n* = 34; Means with the different superscript letters in a column are different significantly (*p* < 0.05) among different CP treatment times.

## Data Availability

The original contributions presented in the study are included in the article, further inquiries can be directed to the corresponding author.

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
