# Peer review of "Effect of Intermittent Low-Pressure Radiofrequency Helium Cold Plasma Treatments on Rice Gelatinization, Fatty Acid, and Hygroscopicity"

_foods, 2024, doi:10.3390/foods13071056_

Round 1

Reviewer 1 Report

Comments and Suggestions for Authors

Dear authors

The main criticisms of your work are:

* This study is a simple exercise.

* The introduction and MM sections are very weak

* The scarce discussion of the presented results.

Since the idea and information provided of this current paper titled “Effect of Intermittent Treatments of Low-Pressure Radio frequency Helium Cold Plasma on Gelatinization parameters, Fatty Acid Profile and Hygroscopic Properties of Rice » are interesting. But, some points which should be addressed in order to improve the quality of the MS

1.     The title should be concise

2.     In the abstract section, comparative data should be introduced

3.     Technical options should be more elucidated, How about the choice of the studied parameters

4.     Authors should stress the novelty of this work

5.     In the introduction section, authors should use recent and suitable references (between 2019 and 2024)

6.     Avoid to use long sentences

7.     Some sentences should more developed with proper references

8.     How about the practical application of CP and its limitations?

9.     If CP affect (negatively or positively) the organoleptic and sensory features?

10.  Authors should discuss several studies linked of this study

11.  Authors should stress the novelty of this work

12.  Please re write the objective, it is not clear

13.  The figure 1 should be deleted

14.  Why authors expose the apparatus of CP, Please develop only the conditions used in this work

15.  L117-129, this part should be concise

16.  Some experiments should be concise

17.  Authors should link all results by a suitable statistical tool (MLR, Peasron, PCA, heat maps, …) since several data was provided

18.  Figure 3, authors add the corresponding superscripts

19.  The SD or SE should be checked in the Table 7, an error of 11% in this value (49/439 in CK at 0 day), the table should be statically verified

20.  The discussion of the results is described very briefly without any thoughts or conclusions as to why this may be so. The results are only described in the form of what came out. It seems to me that this work looks more like a student's final work.

21.  all data should be linked, since this study has several experiments

22.   authors should deeply discuss their results, and compare their results with another recent and suitable works

23.  The link between all data should be established

24. the conclusion   should be improved taking into all remarks and suggestions

Author Response

Reply letter to the reviewer I

1. The title should be concise

Response: Thanks, I have tried to revise as: Effect of Intermittent Low-pressure Radio-frequency Helium Cold Plasma Treatments on Rice Gelatinization, Fatty Acid and Hygroscopicity.

2. In the abstract section, comparative data should be introduced.

Response: Thanks, we revised as: Compared with the untreated (zero times) sample, with an increase in CP treatment times from one to four on rice, the water contact angle and cooking time decreased, while the water absorption rate and freshness index increased, and the pH value remained unchanged.

3. Technical options should be more elucidated, How about the choice of the studied parameters.

Response: Thanks, we revised as: Results support that the effect of low-pressure RF 120W helium CP treatment 20 s on rice grains is perdurable.

4. Authors should stress the novelty of this work.

Response: We sum up as, CP repeatedly treated rice first exhibits the similar monolayer water content and solid surface area of water sorption. Principal component analysis shows that contact angle, pasting parameters and fatty acid profile in milled rice are quite sensitive to CP treatment. Results support that the effect of low-pressure RF 120W helium CP treatment 20 s on rice grains is perdurable, and the improvement of CP intermittent treatments on rice cooking and pasting properties is additive, and the hygroscopic properties of rice was kept.

5. In the introduction section, authors should use recent and suitable references (between 2019 and 2024)

Response: Thanks. The effects of cold plasma systems on cereal food matrix were mainly published during 2012-2019, in recent five years, the researches on CP treated fruits and vegetables are increasing. We renew ref [17-21].

[17] Zhang, J., Du, Q., Yang, Y., Zhang, J., Han, R., Wang, J. Research prgress and future trends of low temperature plasma application in food industry: a review. Molecules. 2023, 28: 4714, doi: 10.3390/molecules28124714.

[18] Usman, I., Afzaal, M., Imran, A., Saeed, F., Afzal, A., Ashfaq, I., Ahah, Y.A., Islam, F., Azam, I., Tariq, I., Ateeq, H., Asghar, A., Farooq, R., Rasheed, A., Shah, M.A. Recent updates and perspectives of plasma in food processing: a review. International Journal of Food Properties. 2023, 26(1): 552-566, doi: 10.1080/10942912.2023.2171052.

[19] Misnal, M.F.I., Redzuan, N., Zainal, M.N.F., Ahmad, N., Ibrahim, K.R., Agun, L. Cold plasma: A potential alternative for rice grain postharvest treatment management in Malaysia. Rice Science, 2022, 29(1): 1-15; doi:10.1016/j.rsci.2021.12.001.

[20] Liu, J.J., Wang, R.L., Chen, Z.T., Li, X.J. Effect of cold plasma treatment on cooking, thermomechanical and surface structural properties of Chinese milled rice. Food and Bioprocess Technology. 2021,14: 866-886; doi:10.1007/s11947-021-02614-1.

[21] Chen, Z.T., Wang, R.L., Li, X.J., Zhu, J., Xu, Y.N., Liu, J.J. Sorption equilibrium moisture and isosteric heat of cold plasma treated milled rice. Innovative Food Science and Emerging Technologies, 2019, 55:35-47, doi:10.1016/j.ifset.2019.05.012.

6. Avoid to use long sentences.

Response: Thanks.

7. Some sentences should more developed with proper references

Response: Thanks.

8. How about the practical application of CP and its limitations?

Response: Clarifying the underying physical and biochemical mechanisms of plasma-cereals interaction is crucial for maximizing desired results.The present study used low-pressure radiofrequency (RF) helium CP (13.56 MHz, 140 Pa, 120 W-20s, the distance between sample and an electrode is 3 cm) to treat 300 g rice sample, the effect on shortening rice cooking time is stable, other senstive organoleptic and biochemical parameters should be examined.

9. If CP affect (negatively or positively) the organoleptic and sensory features?

Response: Thanks, we consider this question in the revised manuscript.

Numerous effects of CP, such as the organoleptic and biomolecular impact, plant cell structure modification, and the microbial impact, have been extensively studied [17-19]. However, there are still many gaps in our understanding of the impact mechanisms of CP on the organoleptic and biomolecular properties of food.These gaps should be thoroughly examined and the potential cumulative effects or alterations can be uncovered to ensure sustained benefits before this technology will be viable for food industry.

10. Authors should discuss several studies linked of this study

Response: Thanks, we compared the acidfication therory, starch gelatinization, pasting vicosity, and polymer swelling (SRC values), as well as IR spectra, fatty acid profile, and thermodynamic properties after CP repeated treatment. SRC values and thermodynamic properties only were treated by the present study.

11. Authors should stress the novelty of this work

Response: We sum up as, CP repeatedly treated rice first exhibits the similar monolayer water content and solid surface area of water sorption. Principal component analysis shows that contact angle, pasting parameters and fatty acid profile in milled rice are quite sensitive to CP treatment. Results support that the effect of low-pressure RF 120W helium CP treatment 20 s on rice grains is perdurable, and the improvement of CP intermittent treatments on rice cooking and pasting properties is additive, and the hygroscopic properties of rice was kept.

12. Please rewrite the objective, it is not clear

Response: Thanks. The present study performed tests just after low-pressure RF helium CP treatment on short-grain milled rice, and repeated the tests several times to demonstrate the changes in gelatinization temperature, fatty acid profile, and hygroscopic properties, with the aim of establishing the effective, safe and reproducible effects of CP technology on food products, and proving the industrial applicability of this technology.

13. The figure 1 should be deleted;

Response: We accept.

14. Why authors expose the apparatus of CP, Please develop only the conditions used in this work

Response: We accept.

15. L117-129, this part should be concise;

Response: Thanks. The rice was repeatedly treated at 120 W for 20 s in the CP apparatus according to the experimental procedure in Figure 1. The untreated (CK) sample was packed into a No.9 valve bag (280 mm length×200 mm width×0.04 mm thickness, Apple Brand, Shanghai China Manufacturing). The CP treated rice (0 h CP) was packaged into a No.9 valve bag and No.11 valve bag (400 mm length×280 mm width×0.04 mm thickness), and stored at RT. After CP treatment at the 48th hour, all the samples were stored at a 4°C refrigerator for 30 days. During the analysis, all the samples were kept in a carton box (21×18×8 cm of length×width height) in the 4°C refrigerator. When CP processing were complete, a portion of rice kernels was stored at a refrigerator, and ca. 200 g of untreated and CP treated rice were respectively milled into rice meal at a Gaosu universal pulverizer (Kewei Yongxing Instrument Co. Ltd., Beijing, China). The pulverizer has 26,000 rpm/min of rotation rate and 100-120 mesh degree of grinding. The rice meal was maintained at -20°C until to analyze.

16. Some experiments should be concise;

Response: Thanks.

17. Authors should link all results by a suitable statistical tool (MLR, Peasron, PCA, heat maps, …) since several data was provided.

Response: Thanks. We adopt PCA analysis for screening the senstive parameter after CP repeatedly treated rice.

18. Figure 3, authors add the corresponding superscripts;

Response: Thanks. We did.

19. The SD or SE should be checked in the Table 7, an error of 11% in this value (49/439 in CK at 0 day), the table should be statically verified

Response: Thanks. In Table 7, the MCPE equation coefficients, A, B, and C are expressed as mean ± standard deviation (SD), number of observations - n= 34, the means with the different superscript letters in a column are different significantly (p < 0.05) among different CP treatment times. The SPSS non-linear regression analysis was used to fit with the measured 34 data sets (temperature, ERH, EMC) on rice kernels in sample chamber. At the same temperature, five samples had similar EMC isotherms, their similar equation coefficients are reasonable.

20. The discussion of the results is described very briefly without any thoughts or conclusions as to why this may be so. The results are only described in the form of what came out. It seems to me that this work looks more like a student's final work.

Response: Thanks.

21. All data should be linked, since this study has several experiments;

Response: Thanks, we adopt principal component analysis.

To the five samples with zero to four CP treatment times, the measured fifty-one indexes were used to principal component analysis. Figure 4 shows the loading plot of principal components after varimax rotation, the five samples were gathered together, suggesting that the effect of CP on milled rice presents difference with CP treatment times, but the difference is not huge. Further make factional scatterplot of principal components of the measured fifty-one parameters (Figure 5), contact angle, pasting parameters (PV, TV, FV, BD, SB) and fatty acid profile (C16:0; C18: 0; C18: 1n9c; C18: 2n6c, total, saturated, unsatu-rated, monounsaturated, polyunsatured FA ) shows dispersed distribution, but other 36 parameteres were gathered together. These reults indicate that contact angle, pasting parameters and fatty acid profile in milled rice are quite sensitive to CP treatment.

22. Authors should deeply discuss their results, and compare their results with another recent and suitable works;

Response: Thanks. In combination of low-pressure radiofrequency CP power (13.56 MHz, 140 Pa) and treatment time, our previous report showed 120 W helium CP treatment 20s on 300 g rice sample had optimal results based on the cooking properties, cooked rice texture, and rice appearance quality, but we worry about how long the effect of CP treatment can be kept. This study further tests this thinking.

23. The link between all data should be established;

Response: Thanks. To the five samples with zero to four CP treatment times, the measured fifty-one indexes were used to principal component analysis.

24. The conclusion   should be improved taking into all remarks and suggestions;

Response: Thanks. The results of principal component analysis are included.

Reviewer 2 Report

Comments and Suggestions for Authors

I send a review of manuscript ID number foods-2722779, of the authors: Ziyi Cao, Xingjun Li, Hongdong Song, Yu Jie and Chang LiuEffect of Intermittent Treatments of Low-Pressure Radio Frequency Helium Cold Plasma on Gelatinization parameters, Fatty Acid Profile and Hygroscopic Properties of Rice.

I think that the manuscript deals with an interesting area of scientific research on the aspect of the impact of cold plasma (CP) technology, to improve rice quality. I think that the authors should make a minor revision.

 Keywords: Please add to the keywords - the word – Fatty Acid Profile or Fatty Acid

  1. Introduction:

 Page 2, lines 77-87; Please clearly formulate and specify the aim of the work (now in a way it resembles a description as in the methodology).

  1. Materials and methods:

Page 3, line 96; should be … shown in Figure 1 (please correct throughout the text).

Page 8, lines 272-273; Please indicate which test was used to determine - Statistical significance?

Page 8, lines 271-273; In this section, please provide information on how many replicates were performed in the submitted studies.

  1. Results

Page 8, lines 285-287 - Please add information under Table 1 regarding - standard deviation (± SD), number of repetitions - n=? and include indication a,b,c,d before the sentence - …Means with the different superscript letters… Please also apply under Tables 2-8.

Page 11, line 377; Please add information under Figure 3 regarding - standard deviation (± SD) and number of repetitions - n=?

 References

In the literature list, the authors should add the missing DOI addresses, next to the relevant literature items, e.g. [1-3, 5], etc. All literature is cited in the text.

Author Response

Reply letter to reviewer II

I think that the manuscript deals with an interesting area of scientific research on the aspect of the impact of cold plasma (CP) technology, to improve rice quality. I think that the authors should make a minor revision.

Response: Thanks.

Keywords: Please add to the keywords - the word – Fatty Acid Profile or Fatty Acid.

Response: Thanks. We did.

Introduction:

Page 2, lines 77-87; Please clearly formulate and specify the aim of the work (now in a way it resembles a description as in the methodology).

Response: We revised as: The present study performed tests just after low-pressure RF helium CP treatment on short-grain milled rice, and repeated the tests several times to demonstrate the changes in gelatinization temperature, fatty acid profile, and hygroscopic properties, with the aim of establishing the effective, safe and reproducible effects of CP technology on food products, and proving the industrial applicability of this technology.

Materials and methods:

Page 3, line 96; should be … shown in Figure 1 (please correct throughout the text).

Response: Thanks, we adopt Figure 1.

Page 8, lines 272-273; Please indicate which test was used to determine - Statistical significance?

Page 8, lines 271-273; In this section, please provide information on how many replicates were performed in the submitted studies.

Response: Except for parallel samples for the determinination of EMC/ERH data, at least three replicates were tested on each rice sample for physico-chemical and rice quality pa-rameters. SPSS software (Version 17.0 [21]) was used to analyze data. One-way analysis of variance (ANOVA) and Duncan's new multiple-range test was used for comparing multiple of means. Statistical significance was stated at p < 0.05. Data reduction and factor method was used for principal components analysis and to make factional scatterplot.

Results

Page 8, lines 285-287 - Please add information under Table 1 regarding - standard deviation (± SD), number of repetitions - n=? and include indication a,b,c,d before the sentence - …Means with the different superscript letters… Please also apply under Tables 2-8.

Response: Thanks, we did.

Page 11, line 377; Please add information under Figure 3 regarding - standard deviation (± SD) and number of repetitions - n=?

Response: Thanks, we did.

References

In the literature list, the authors should add the missing DOI addresses, next to the relevant literature items, e.g. [1-3, 5], etc. All literature is cited in the text.

Response: Thanks, we did.

Round 2

Reviewer 1 Report

Comments and Suggestions for Authors

accept